# Effectiveness in the Block by Honokiol, a Dimerized Allylphenol from *Magnolia Officinalis*, of Hyperpolarization-Activated Cation Current and Delayed-Rectifier K^+^ Current

**DOI:** 10.3390/ijms21124260

**Published:** 2020-06-15

**Authors:** Ming-Huan Chan, Hwei-Hsien Chen, Yi-Ching Lo, Sheng-Nan Wu

**Affiliations:** 1Institute of Neuroscience, National Chengchi University, Taipei 11605, Taiwan; minghuan@nccu.edu.tw (M.-H.C.); hwei@nhri.org.tw (H.-H.C.); 2Center of Neuropsychiatric Research, National Health Research Institutes, Miaoli County 35053, Taiwan; 3Department of Pharmacology, Kaohsiung Medical University, Kaohsiung 80708, Taiwan; yichlo@kmu.edu.tw; 4Department of Physiology, National Cheng Kung University Medical College, Tainan 70101, Taiwan; 5Institute of Basic Medical Sciences, National Cheng Kung University Medical College, Tainan 70101, Taiwan; 6Department of Medical Research, China Medical University Hospital, China Medical University, Taichung 40402, Taiwan

**Keywords:** honokiol, hyperpolarization-activated cation current, current kinetics, pituitary cell, olfactory neuron

## Abstract

**Background:** Honokiol (HNK), a dimer of allylphenol obtained from the bark of *Magnolia officinalis* was demonstrated to exert an array of biological actions in different excitable cell types. However, whether or how this compound can lead to any perturbations on surface–membrane ionic currents remains largely unknown. Methods: We used the patch clamp method and found that addition of HNK effectively depressed the density of macroscopic hyperpolarization-activated cation currents (*I*_h_) in pituitary GH_3_ cells in a concentration-, time- and voltage-dependent manner. By the use of a two-step voltage protocol, the presence of HNK (10 μM) shifted the steady-state activation curve of *I*_h_ density along the voltage axis to a more negative potential by approximately 11 mV, together with no noteworthy modification in the gating charge of the current. Results: The voltage-dependent hysteresis of *I*_h_ density elicited by long-lasting triangular ramp pulse was attenuated by the presence of HNK. The HNK addition also diminished the magnitude of deactivating *I*_h_ density elicited by ramp-up depolarization with varying durations. The effective half-maximal concentration (IC_50_) value needed to inhibit the density of *I*_h_ or delayed rectifier K^+^ current identified in GH_3_ cells was estimated to be 2.1 or 6.8 μM, respectively. In cell-attached current recordings, HNK decreased the frequency of spontaneous action currents. In Rolf B1.T olfactory sensory neurons, HNK was also observed to decrease *I*_h_ density in a concentration-dependent manner. **Conclusions:** The present study highlights the evidence revealing that HNK has the propensity to perturb these ionic currents and that the hyperpolarization-activated cyclic nucleotide-gated (*HCN*) channel is proposed to be a potential target for the in vivo actions of HNK and its structurally similar compounds.

## 1. Introduction

Honokiol (HNK), a hydroxylated biphenyl compound obtained from *Magnolia officinalis* and from other species of the family Magnoliaceae, has been used in traditional Asian medicines (Houpo, Hou p’u, or Saiboku-tu(o)) [1]. HNK is recognized to be a potential natural compound that has been demonstrated to exert multiple effects on different cellular processes in various cancer models [2,3,4,5,6]. Previous studies have also revealed the effectiveness of this compound in modulating the functional activities of neuroendocrine or endocrine cells. For example, an earlier study has demonstrated that HNK could induce cell cycle arrest and program cell death in vitro and in vivo in human thyroid neoplastic cells [3]. Several investigations have also reported the ability of magnolia bark or HNK to modify the secretion of catecholamines from adrenal chromaffin cells [7,8]. Moreover, HNK was noted to exert antidepressant effects by normalizing limbic hypothalamic-pituitary-adrenal axis [9,10].

The bark of *M. obovata* Thunberg has been prescribed in many Chinese traditional medicines used to treat mental and neurological disorders such as anxiety and depression [1]. The extract of magnolia bark with ester has been reported to show depressant actions on the central nervous system, such as sedation, loss of righting reflex, and the depression of drug-induced convulsion [11,12,13]. Previous reports have revealed that the mixture of HNK and magnolol (an isomer of HNK) possessed antidepressant-like properties, which had an effect on the behaviors of rodents with chronic mild stress [9,10]. *M. officinalis* was demonstrated to reduce the long-term effects of status epilepticus induced by kainic acid [13]. Nanosomes used for the encapsulation of HNK were also recently reported to improve the feasibility of HNK for its intravenous administration tailored against autoimmune encelphalomyelitis [14,15]. Additionally, at the cellular level, HNK or magnolol has been shown to induce Ca^2+^ mobilization in cortical neurons and in neuroblastoma cells [16], as well as to suppress glutamate-evoked Ca^2+^ influx in rat cerebellar granule cells [17].

Notably, HNK has been previously reported to suppress NMDA receptor-mediated nociception as well as mGluR5-mediated response [18]. Several studies have described the ability of HNK to exert anti-hyperalgesic properties in inflammatory pain models [11,19,20,21]. This compound or magnolol could also attenuate prostaglandin E2-induced thermal hyperalgesia for up to 120 min following pain induction [18]. Furthermore, it was recently demonstrated to ameliorate the postoperative cognitive impairment induced by surgery/sevoflurane, possibly through its inhibitory effect on the activation of the NLR family pyrin domain containing 3 (*NLRP3*) inflammasome in the hippocampus [22].

The hyperpolarization-activated cation current (termed *I*_h_ or funny current [*I*_f_]) has been recognized as a key determinant of repetitive electrical activity in heart cells and in a variety of central neurons, and endocrine or neuroendocrine cells [23,24,25,26,27,28,29,30]. This current with its unusually slow voltage-dependent activation kinetics is a mixed inwardly directed Na^+^/K^+^ current, which is sensitive to blocking by CsCl, ivabradine or zatebradine [31,32,33,34]. The activation of this current can have the propensity to cause the resting potential to be depolarized and then to reach the threshold required for the generation of action potential; as a result, it is thereby able to influence pacemaker activity and impulse propagation [27,28,29,35]. Additionally, the slow kinetics of *I*_h_ in response to a long hyperpolarization clamp can also produce long-lasting activity-dependent perturbations in the membrane excitability of many excitable cell types [28,34]. This type of current is recognized to be carried by the channels encoded by a family of the hyperpolarization-activated cyclic nucleotide-gated (*HCN*) gene. The *HCN* channels belong to the superfamily of voltage-gated K^+^ channels and cyclic nucleotide-gated channels, and they are tetrameric proteins in which four individual subunits (i.e., homotetrameric or heterotetrameric subunits) are required to constitute a functional channel, the activity of which could be linked to pain sensation [27,32,34,35,36].

Previous studies have revealed the effectiveness of magnolol in stimulating the activity of large-conductance Ca^2+^-activated K^+^ channels in tracheal smooth muscle cells or in the small intestine [37,38]. Magnolol was also previously shown to inhibit voltage-gated Na^+^ and K^+^ channels in NG108–15 neuroblastoma cells and in in freshly isolated mouse dorsal root ganglion neurons [39,40]. However, how HNK or other structurally similar compounds are capable of interacting with these *HCN* channels to alter the density and gating of *I*_h_ or other types of ion currents in different excitable cell types still largely remains elusive.

In taking into account the above descriptions, the aim of this work was to investigate whether HNK could exert any modifications on the density and gating of *I*_h_ identified in pituitary tumor (GH_3_) cells and in Rolf B1.T olfactory neurons. Notably, findings from the present observations provide the evidence to show, first and foremost, that in addition to its block on delayed-rectifier K^+^ current (*I*_K(DR)_), the presence of HNK can effectively interact with the *HCN* channel to depress *I*_h_ density in a concentration-, time-, and voltage-dependent manner. Under cell-attached current recordings, its presence was also found to decrease the frequency of spontaneous action currents in GH_3_ cells.

## 2. Results

### 2.1. Effect of HNK on the Density of Hyperpolarization-Activated Cation Current (I_h_) in GH_3_ Cells

The whole-cell configuration of the standard patch clamp technique was used to investigate the effect of HNK or other related compounds on macroscopic ionic currents. In these experiments, as cells were bathed in Ca^2+^-free Tyrode’s solution, the applied long-lasting step hyperpolarization was noted to induce an inwardly directed current with the slowly activating and deactivating properties, which has been previously recognized as the hyperpolarization-activated cation current, namely, *I*_h_ or funny current (*I*_f_) (Figure 1A,B) [25,26,30,35,41,42,43]. Within 1 min of exposing the GH_3_ cell to HNK (3 or 10 μM), the *I*_h_ density responding to such membrane hyperpolarization was robustly reduced. For example, when cells were hyperpolarized for 2 s from −40 to −120 mV, the presence of HNK (3 μM) notably reduced the density of the late inward current at the end of 2 s hyperpolarization from 9.4 ± 0.9 to 6.8 ± 0.7 pA/pF (*n* = 7, *p* < 0.05); meanwhile, the density of deactivating *I*_h_ upon the return to −40 mV was detected to decline from 7.9 ± 0.8 to 3.9 ± 0.6 pA/pF (*n* = 7, *p* < 0.05). As shown in Figure 1C, the activation time constant (τ_act_) of *I*_h_ density during long membrane hyperpolarization also increased. This depressant effect was readily reversed upon the removal of the compound.

The relationship between the HNK concentration and the relative density of *I*_h_ is illustrated in Figure 1B. The presence of HNK was found to depress the density of *I*_h_ during long membrane hyperpolarization in a concentration-dependent manner. From the analysis of the modified Hill equation, the half-maximal concentration required for inhibitory effect of HNK on the density of hyperpolarization-induced *I*_h_ was estimated to be 2.1 μM; and, HNK at a concentration of 30 μM almost fully decreased current density.

The mean current density-voltage relationships for *I*_h_ taken with or without the addition of HNK (3 μM) displaying non-ohmic behavior were established and are plotted in Figure 1D. As cells were exposed to HNK, the *I*_h_ densities biophysically characterized by an inwardly rectifying property, were depressed throughout the entire voltage clamp step applied. For example, HNK (3 μM) decreased whole-cell conductance of *I*_h_ linearly measured at the voltages between −120 and −100 mV from 0.38 ± 0.03 to 0.09 ± 0.01 nS/pF (*n* = 7, *p* < 0.05); moreover, washout of the compound, the *I*_h_ conductance was returned to 0.30 ± 0.02 nS/pF (*n* = 7, *p* < 0.05).

### 2.2. Modification of the Steady-State Activation Curve of I_h_ Produced by the Presence of HNK

To characterize the inhibitory effect of HNK on *I*_h_ density in GH_3_ cells, we further wanted to determine its possible modifications on the steady-state activation curve of *I*_h_ density. Figure 2 illustrates the voltage-dependent activation curve of *I*_h_ density collected with or without the HNK (10 μM) addition. In this set of experiments, we designed and then applied a two-step voltage pulse protocol generated through digital to analog conversion. In other words, a 2-s conditioning pulse was delivered to various potentials to precede the 2-s hyperpolarizing pulse of −120 mV from a holding potential of −40 mV. During the recordings, the intervals between two sets of voltage pulses were set to be approximately 20 min to ensure complete recovery of *I*_h_. The relationships between the conditioning potentials and the normalized densities (I density/I_max_ density) in the absence and presence of 10 μM HNK were established and then fitted with the goodness of fit by a modified Boltzmann relation as elaborated in Materials and Methods. In control, *V*_1/2_ = −84.1 ± 2.1 mV (*n* = 7), *q* = 4.6 ± 0.8 *e* (*n* = 7), whereas during exposure to 10 μM HNK, *V*_1/2_ = −95.2 ± 2.2 mV (*n* = 7, *p* < 0.05), *q* = 4.7 ± 0.8 *e* (*n* = 7, *p* > 0.05). It is conceivable, then, that the presence of HNK not only reduced the maximal conductance of *I*_h_, but also significantly shifted the activation curve in parallel to a hyperpolarized potential by approximately 11 mV. Conversely, we were unable to detect any significant change in the gating charge of the curve acquired between the absence and presence of 10 μM HNK. The results were interpreted to indicate that the addition of HNK is capable of altering the steady-state activation curve of *I*_h_ density in GH_3_ cells.

### 2.3. Effect of HNK on Voltage-Dependent Hysteresis of I_h_ Elicited in Response to Long Triangular Ramp Pulse

It has been previously demonstrated that the voltage-dependent hysteresis of *I*_h_ density can exert a high influence on electrical behaviors such as action potential firing [41,44,45,46]. As such, we further explored whether HNK could perturb the voltage hysteresis of this current density identified in GH_3_ cells. In this stage of experiments, we exploited a long-lasting triangular ramp pulse with a duration (i.e., ±0.11 V/s) for the measurement of the hysteretic properties, as whole-cell configuration was achieved. It is evident from Figure 3 that the trajectory of *I*_h_ density in response to the upsloping (i.e., depolarization from −150 to −40 mV) and downsloping (hyperpolarizing from −40 to −150 mV) ramp pulse as a function of time was noticeably distinguishable between these two limbs. The current density responding to the upsloping limb of long-lasting triangular ramp was considerably larger than that by the downsloping limb, that is, there is a conceivable voltage-dependent hysteresis for the current density in terms of the relationship of *I*_h_ density versus membrane potential. As the ramp speed was reduced, the hysteretic degree for *I*_h_ density progressively increased. Interestingly, as the examined cell was exposed to 3 μM HNK, *I*_h_ density during the upsloping limb of the triangular ramp was detected to increase to a greater extent than that which was measured from the downsloping limb. For example, in the controls, *I*_h_ density at the level of −110 mV by the upsloping or downsloping limb of the triangular ramp pulse was measured to be 9.7 ± 0.2 and 1.9 ± 0.1 pA/pF (*n* = 8, *p* < 0.05), respectively, the values of which were found to differ noticeably in current densities between these two limbs. Moreover, as GH_3_ cells were exposed to 3 μM HNK, the density of the triangular ramp-induced forward current at the same level of membrane potential was considerably decreased to 5.2 ± 0.2 pA/pF (*n* = 8, *p* < 0.05), while that of the backward current (1.9 ± 0.1 pA/pF, *p* > 0.05) was not significantly changed.

We further characterized and quantified the degree of voltage-dependent hysteresis of Ih density on the basis of the differences in areas under the curves (indicated in shaded area) in the forward (upsloping) and reverse (downsloping) directions, as denoted by the dashed arrows in Figure 3A [41]. It was seen that, for *I*_h_ in GH_3_ cells, the degree of voltage hysteresis increased with slower ramp speed, and that the presence of HNK led to a conceivable reduction in the amount of such hysteresis. Figure 3B depicts a summary of the data revealing the effects of HNK and HNK plus oxaliplatin on the hysteretic area under the curve between forward and backward current traces. For example, apart a from reduction in *I*_h_ density magnitude, the addition of 3 μM HNK decreased the area by 65%, which was elicited during the long-lasting triangular voltage ramp; moreover, the subsequent application of 10 μM oxaplatin increased the area 1.4-fold. Oxaliplatin, a chemotherapeutic agent, was recently demonstrated to activate *I*_h_ [30,36,41].

In another separate set of experiments, we applied an inverted triangular ramp pulse to the cell and determined the extent of voltage hysteresis of *I*_h_ density identified in GH_3_ cells. As illustrated in Figure 3C, the magnitude of hysteretic Δarea in response to the forward downsloping and backward upsloping directions became greater compared to that in Figure 3A. In accordance with the above-described observations revealed in Figure 3A, the area under the curve of forward and backward direction was evidently decreased in the presence of 3 μM HNK. It is noticeable from these experimental data that cell exposure to HNK was able to modify the extent of the voltage-dependent hysteresis of *I*_h_ density in these cells.

### 2.4. Effect of HNK on Deactivating I_h_ Elicited upon Return to Membrane Depolarization with Varying Duration

Previous studies have demonstrated that the magnitude of *I*_h_ in electrically excitable cells might greatly influence the rising phase of pacemaker potential or abrupt spike depolarizations [24,34,44,47,48]. Thus, next, we intended to determine how HNK exerted any perturbations on the deactivating *I*_h_ in response to the upsloping ramp pulse from −120 to −40 mV with varying durations. As revealed in Figure 4A,B, upon the return to −40 mV, as the slope of the ramp-up pulse was slowed, the peak density of deactivating *I*_h_ became exponentially decreased with a time constant of 126 ± 8 ms (*n* = 9). An abrupt hump occurring during the upsloping ramp was evidently found in the time course of *I*_h_ density deactivation. This property can account for the inwardly rectifying phenomenon (i.e., anomalous rectification) in the instantaneous current versus voltage relationship during the rising event of pacemaker potential [49,50]. However, as GH_3_ cells were exposed to 3 μM HNK, the peak density of the current progressively decreased, whereas the decaying time course of the current was exponentially prolonged with a time constant of 426 ± 16 ms (*n* = 9, *p* < 0.05). For example, as the duration of the upsloping ramp was set at 100 ms (i.e., slope = 0.8 V/s), the addition of 3 μM HNK noticeably decreased peak density by 32 ± 7% from 12 ± 0.7 to 8.2 ± 0.6 pA/pF (*n* = 9, *p* < 0.05). As the ramp-up duration increased from 100 to 400 ms, the rising rate of deactivating *I*_h_ was conceivably reduced to 0.023 ± 0.001 (pA/pF)/ms from a control of 0.115 ± 0.003 (pA/pF)/ms (*n* = 9, *p* < 0.05). Meanwhile, the addition of 3 μM HNK further diminished the rising *I*_h_ rate at the ramp-up duration of 100 or 400 ms to 0.0776 ± 0.002 (pA/pF)/ms (*n* = 9, *p* < 0.05) or 0.0148 ± 0.001 (pA/pF)/ms (*n* = 9, *p* < 0.05), respectively. This occurrence thus reflected the fact that, as the duration of upsloping ramp pulse increased, the density of deactivating *I*_h_ would be exponentially decreased, and that the presence of HNK decreased the current magnitude in a time-dependent fashion.

### 2.5. Inhibitory Effect of HNK on Delayed-Rectifier K^+^ Current (I_K(DR)_) in GH_3_ Cells

In a separate set of experiments, we also assessed the effect of HNK on *I*_K(DR)_ density in these cells. The major determinants of *I*_K(DR)_ in GH_3_ cells are the voltage-gated K^+^ channels from K_V_3.1-K_V_3.2 types and these were thought to play a role in determining membrane excitability and the repolarization of action potential [28]. To measure *I*_K(DR)_, we bathed cells in Ca^2+^-free Tyrode’s solution which contained tetrodotoxin (1 μM) and CdCl_2_ (0.5 mM), and the recording pipette was filled with a K^+^-containing solution. Figure 5 shows that HNK can depress the density of *I*_K(DR)_ in a concentration-dependent fashion. In these recordings, each cell was depolarized for 1 s from −50 mV to various potentials ranging between −50 and +50 mV, and current densities were measured at the end of depolarizing step commands. Figure 5A,B illustrates the mean current density versus voltage relationships of *I*_K(DR)_ obtained in the control and during exposure to 10 μM HNK. For example, at the level of +50 mV, the presence of 10 μM HNK decreased *I*_K(DR)_ density from 28.7 ± 2.4 to 21.4 ± 2.1 pA/pF (*n* = 8, *p* < 0.05) and, after the washout of the compound, current density was returned to 28.4 ± 2.1 pA/pF (*n* = 8, *p* < 0.05). Meanwhile, as cells were exposed to 10 μM HNK, whole-cell conductance of *I*_K(DR)_ measured at voltages ranging between +30 and +50 mV was significantly decreased to 0.44 ± 0.01 nS/pF from a control of 0.58 ± 0.02 nS/pF (*n* = 7, *p* < 0.05). Despite its depressant action on *I*_K(DR)_ density, neither the activation nor inactivation time course of the current density in response to membrane depolarization was notably altered in the presence of HNK. The relationship between the HNK concentration and the relative density of *I*_K(DR)_ was also established and is plotted in Figure 5C. Fitting the concentration–response curve with the modified Hill equation, as described in **Materials and Methods**, thereafter gave a half-maximal concentration (IC_50_) of 6.8 μM and a slope coefficient of 1.1. HNK, at a concentration of 100 μM, decreased current density by about 35%.

### 2.6. Inhibitory Effect of HNK on Spontaneous Action Currents (ACs) in GH_3_ Cells

In another set of experiments, we investigated the effects of HNK on spontaneous action currents (ACs) in these cells. Cells were immersed in normal Tyrode’s solution containing 1.8 mM CaCl_2_. When cell-attached current recordings were firmly achieved, we were able to detect spontaneous ACs. As cells were exposed to HNK at a concentration of 1 or 3 μM, a progressive decrease in the frequency of ACs was observed (Figure 6). It is possible, therefore, that the HNK-induced decrease in spontaneous ACs is mediated largely by inhibition actions on I_h_, as described above.

### 2.7. Inhibitory Effect of HNK on I_h_ Density Identified in Rolf B1.T Olfactory Neurons

In a final set of recordings, we further examined if the presence of HNK could produce any effects on *I*_h_ density inherently in sensory neurons such as Rolf B1.T cells. The results showed that the addition of HNK effectively decreased *I*_h_ density in a concentration-dependent manner (Figure 7A,B). For example, as cells were exposed to 10 μM HNK, current density measured at the level of −120 mV was significantly decreased to 1.3 ± 0.1 pA/pF from a control of 5.7 ± 0.7 pA/pF (*n* = 8, *p* < 0.05) and, after washout of HNK, current density returned to 5.6 ± 0.7 pA/pF (*n* = 8, *p* < 0.05). Meanwhile, ivabradine (3 μM) or linalool (10 μM) was effective at depressing *I*_h_ density measured at −120 mV. Ivabradine was recognized as an inhibitor of *I*_h_ [32,33,34,41,42], while linalool is an aromatic plant-derived monoterpene alcohol [51]. Moreover, as illustrated in Figure 6B, in the continued presence of HNK, the subsequent application of oxaliplatin (10 μM) significantly counteracted the HNK-mediated inhibition of *I*_h_ density in these cells. It is clear from these data that the presence of this compound was effective at decreasing the *I*_h_ density identified in this type of olfactory neurons.

## 3. Discussion

One factor that is pertinent to findings in the present study is that, unlike its action on *I*_K(DR)_ density, the presence of HNK has the propensity to interact with *HCN* channels to decrease the density of *I*_h_ as well as to slow down the activating and deactivating rate of the current density during long hyperpolarizing steps. In other words, although HNK was effective at depressing the density of both *I*_h_ and *I*_K(DR)_, instead of being decreased, the activating and deactivating time courses of *I*_h_ density elicited by long-step membrane hyperpolarization became slower. The reason for this discrepancy is currently unknown; however, the HNK action seems to be cell type related and dependent on the specific types of K_V_ or *HCN* channels present. This also probably reflects changes in the location of the binding site(s) produced by differences in the channel sequence of *HCN* and K_V_ channels. Regardless of the detailed ionic mechanism of its inhibitory actions, this study served as a platform for us to propose that HNK is a slow on–off blocker of *I*_h_ density elicited in response to long step hyperpolarization. The block of *I*_h_ density caused by HNK is able to retard the generation of pacemaker potential, which may render excitable cells to decrease the rhythmic firing of action potentials.

In this study, the steady-state activation curve of *I*_h_ density observed in GH_3_ cells was apparently shifted along the voltage axis toward a negative voltage in the presence of HNK. However, the lack of an effect on the gating charge of the current was detected in its presence, reflecting that the HNK action on the channel might act as a gate to open the channel, but not, instead, act on the region that senses the transmembrane potential, since there was no change in the gating charges that were allowed to cross the transmembrane electric field during *I*_h_ density activation. Additionally, as the duration of the rising phase upon the return from membrane potential of −120 to −40 mV was increased, the magnitude of HNK-induced block of *I*_h_ density became progressively raised. Taken together, this means that the sensitivity of neurons or pituitary cells to HNK could depend not simply on the HNK concentration achieved, but also the pre-existing level of the resting potential, the firing behavior, or their combination.

The gating charge of *HCN* has been estimated to be around 7.4 *e* [52], a value which is slightly higher than that demonstrated here. The reason for such a discrepancy is currently unclear; however, it could be due to the different maneuvers used. The estimate employed in this study was based on the quasi-steady-state activation curve of *I*_h_ density by means of a long-lasting two-step voltage protocol. It also must be noted that the macroscopic *I*_h_ could be a mixture of several channel currents (i.e., *HCN*x-encoded currents) in GH_3_ cells.

The physiological importance of *I*_h_ (e.g., *HCN*x-encoded currents) has been recognized in an array of excitable cell types that include cardiac myocytes, sensory neurons and lactotrophs [24,27,32,36,53]. In fact, in this study, we also found that *I*_h_ density, which was sensitive either to block by ivabradine and HNK, or to stimulation by oxaliplatin, was present in Rolf B1.T olfactory neurons. It will thus be of importance to evaluate whether HNK directly depresses the *I*_h_ in cardiac myocytes (e.g., sinoatrial cells), since this effect may lead to decreased rhythmic activity in the heart [23,24,33,50,53].

The IC_50_ value required for the HNK-mediated inhibition of *I*_h_ or *I*_K(DR)_ density demonstrated in this study was 2.1 or 6.8 μM, respectively. Effective IC_50_ used for the HNK-mediated inhibition of *I*_K(DR)_ density was lower than that for the decrease in *I*_K(DR)_ in dorsal root ganglion neurons or neuroblastoma cells [39,40]. Due to the low values of IC_50_ in the blocking of both *I*_h_ and *I*_K(DR)_ densities, such inhibitory actions may thus synergistically act to influence the functional activity of these cells in vivo. There might be a pertinent link between the actions of HNK on different types of electrically excitable cells (e.g., GH_3_ cells) and its observed effects on membrane ion channels. Further investigations are needed to find out whether HNK can affect either *I*_h_ existing in a variety of cells or different types of *I*_h_ (e.g., *HCN*x-encoded currents) [29,45].

The voltage-dependent hysteresis of *I*_h_ density is thought to serve a role in influencing the electrical behavior of electrically excitable cells including GH_3_ cells. In keeping with previous observations [30,41,44,45,46], the *I*_h_ density inherent in GH_3_ cells was apparently described either to undergo a hysteretic perturbation in situations where either its voltage dependence, or a significant mode shift in which the voltage sensitivity in the gating charge movements of the current, occurs. In other words, the magnitude of *I*_h_ density can depend on the previous state of the *HCN* channel [44,45]. In this study, we also examined the possible modifications of HNK on such a non-equilibrium property of *I*_h_ density inherent in GH_3_ cells. Our results demonstrated that the presence of this compound was apparently capable of diminishing the hysteresis involved in the voltage-dependent activation of *I*_h_ density. Moreover, with the continued presence of HNK, the subsequent addition of oxaliplatin could attenuate HNK-mediated reduction in the Δarea of the voltage-dependent hysteresis of the current density. Additionally, although the voltage ranges in which *I*_h_ activation occurs, either in control conditions or after HNK treatment, appear to fall outside of the values of the membrane in a neuron, a small fraction of *I*_h_ is tonically activated at rest [54].

The kinetic effects that form the basis of Figure 3 and Figure 5 also need to be clarified. The hysteresis drawn appears to be generated mainly by what one would term ‘residual activation’. In other words, the deactivation tails of *I*_h_ had a very slow time course, and a sigmoidal time course in *I*_h_ activation was observed. Therefore, it is likely that the presence of HNK could decrease the residual activation of *I*_h_ responding to long-lasting membrane hyperpolarization.

Previous studies have demonstrated the effectiveness of HNK in blocking store-operated Ca^2+^ entry in CHO cells expressing the M_3_ muscarinic receptor, possibly via a mechanism through its binding to muscarinic receptors [16,55]. However, in the present observations, the subsequent addition of atropine (10 μM), still in the continued presence of HNK (10 μM), was found to produce minimal modifications on HNK-mediated inhibition of *I*_h_ in GH_3_ cells. Consequently, it seems unlikely that the block of *I*_h_ density produced by HNK resulted from its direct interaction with its binding to muscarinic receptors and the accompanied changes in intracellular signaling transduction.

A recent study has shown the ability of HNK to suppress transient receptor potential-V1 (TRPV1) and purinergic-2Y (P2Y) nociceptors in mice with third degree burns [21]. One might expect that the TRP superfamily of cation channels in GH_3_ cells [28] could be modified by the presence of HNK. It is important to note, however, that, being distinguishable from those of *I*_h_ density, the biophysical properties of TRPV-mediated currents exhibit themselves as relatively time or voltage independent. The presence of neither an outwardly nor inwardly rectifying property was clearly found in this type of currents. In this regard, it is unlikely that *I*_h_ density depressed by HNK in GH_3_ or Rolf B1.T cells is predominantly mediated by its changes in the activity of TRPV or other TRP-like channels.

Collectively, the present study shows evidence that the HNK presence is effective at inhibiting *I*_h_ and *I*_K(DR)_ densities in pituitary GH_3_ cells. The *I*_h_ density identified in Rolf B1.T olfactory neurons was also subject to blocking by HNK or linalool (a constituent of the essential oil). The inhibition of these ionic currents was noted to be rapid in onset and is hence likely to be responsible for their modulatory action on functional activities in sensory neurons such as Rolf B1.T cells. Such inhibitory actions could lead to modifications in the firing behavior of electrically excitable cells, hence altering neuronal function. It must be mentioned that the electrophysiological effects of HNK demonstrated herein tend to be upstream of its possible effects on either the signal transducer and activator of transcription 3 (STAT3) or nuclear factor erythroid 2 (Nrf2) signaling pathway [2,4,20], the activation of the NLRP3 inflammasome [22], or the regulation of oncogenic transcription factor (forkhead box M1 [FOXM1]) [5]. Additionally, from the pharmacokinetic studies of HNK and magnolol, following oral administration, HNK can readily pass across the blood–brain barrier [12,56,57]. The extent to which the HNK-induced inhibition of ion channels contributes to anti-inflammatory or antinociceptive actions [11,15,18,20,21] is yet to be investigated.

## 4. Materials and Methods

### 4.1. Chemicals, Drugs and Solutions

Honokiol (HNK; Hou p’u, Hòupò, C_18_H_18_O_2_, 2-(4-hydroxy-3-prop-2-enyl-phenyl)-4-prop-2-enyl-phenol, https://pubchem.ncbi.nlm.nih.gov/compound/honokiol), sodium *N*,*N*-diethyldithiocarbamate, ivabradine, atropine, linalool, and tetrodotoxin were acquired from Sigma-Aldrich (Merck Ltd., Taipei, Taiwan), and oxaliplatin was from Sanofi (Taipei, Taiwan). Part of HNK was kindly provided by Dr. Chien-Chich Chen, National Institute of Chinese Medicine, Taipei, Taiwan, while chlorotoxin was provided by Professor Dr. Woei-Jer Chuang (Department of Biochemistry, National Cheng Kung University Medical College). Unless specified otherwise, culture media, horse or fetal calf serum, L-glutamine, trypsin/ethylenediaminetetraaceetic acid (EDTA), penicillin–streptomycin and fungizone were acquired from HyClone^TM^ (Fisher Scientific, Taipei, Taiwan), whereas other chemicals such as CdCl_2_, ethylene glycol-bis(β-aminoethyl ether)-*N*,*N*,*N*’,*N*’-tetraacetic acid (EGTA) and 4-(2-hydroxyethyl)-1-piperazineethanesulfonic acid (HEPES), were commercially and available and of reagent grade.

The HEPES-buffered normal Tyrode’s solution used in this study contained (in mM): NaCl 136.5, KCl 5.4, CaCl_2_ 1.8, MgCl_2_ 0.53, glucose 5.5, and HEPES 5.5 adjusted with NaOH to pH 7.4. To record *I*_h_, *I*_K(DR)_ or ACs, we filled the patch electrode by using the following solution (composition in mM): K-aspartate 130, KCl 20, KH_2_PO_4_ 1, MgCl_2_ 1, Na_2_ATP 3, Na_2_GTP 0.1, EGTA 0.1, and HEPES 5 adjusted with KOH to pH 7.2. In order to avoid the contamination of the Cl^−^ current [53,58], Cl^−^ ions inside the pipette solution were replaced with aspartate. All solutions were prepared using demineralized water from a Milli-Q water purification system (Merck, Ltd., Taipei, Taiwan). The pipette solution and culture medium were filtered on the day of use with an Acrodisc^®^ syringe filter and a 0.2-μm Supor^®^ membrane (Bio-Check; New Taipei City, Taiwan).

### 4.2. Cell Preparations

GH_3_, a clonal cell line derived from a rat prolactin-secreting pituitary tumor, was obtained from the Bioresources Collection and Research Center (BCRC-60015; Hsinchu, Taiwan). Briefly, cells were cultured in Ham’s F-12 medium supplemented with 15% heat-inactivated horse serum (*v*/*v*), 2.5% fetal calf serum (*v*/*v*) and 2 mM L-glutamine in a humidified environment of 5% CO_2_/95 air [59]. To promote cell differentiation, GH_3_ cells were transferred to a serum-free, Ca^2+^-free medium. Rolf B1.T olfactory neurons were maintained in Dulbecco’s modified Eagle’s medium supplemented with 2 mM L-glutamine and 10% fetal bovine serum (*v*/*v*) [60,61]. Under our experimental conditions, cells remained 80–90% viable for at least two weeks. Subcultures were obtained by trypsinization (0.025% trypsin solution (HyClone^TM^) containing 0.01% sodium *N*,*N*-diethyldithiocarbamate and EDTA).

### 4.3. Electrophysiological Measurements

Shortly before the experiments, an aliquot of cell suspension that contained GH_3_ cells or Rolf B1.T olfactory neurons, were gently harvested and transferred immediately to a home-made recording chamber which was firmly positioned on the stage of a CKX-41 inverted microscope (Olympus; YuanLi, Kaohsiung, Taiwan). Cells were immersed at room temperature (22–25 °C) in normal Tyrode’s solution, the composition of which is detailed above. Patch clamp recordings under the whole-cell mode were applied with either an RK-400 (Biol-Logic, Claix, France) or an Axopatch-200B amplifier (Molecular Devices; Bestgen Biotech, New Taipei City, Taiwan) [41,59]. Patch electrodes with tip resistances of 3–5 MΩ were made of Kimax-51 glass capillaries (#34500; Kimble; Dogger, New Taipei City, Taiwan) on either a PP-83 vertical puller (Narishige; Major Instruments, New Taipei City, Taiwan) or a *p*-97 horizontal puller (Sutter, Novato, CA), and then fire-polished with MF-83 microforge (Narishige). Spontaneous ACs were measured by using cell-attached voltage clamp recordings, and the potential was held at the level of the resting potential (around −70 mV) [62].

### 4.4. Data Recordings

The signals, comprising voltage and current tracings, were stored online at 10 kHz in an ASUS VivoBook Flip-14 touchscreen laptop computer (TP412U; Taipei, Taiwan) equipped with the Digidata 1440A interface (Molecular Devices). The latter device was used for efficient analog to digital/digital to analog conversion. During the recordings, the data acquisition system was driven by pCLAMP 10.7 software (Molecular Devices) run under Windows 10 (Redmond, WA, USA), and the signals were simultaneously monitored on a liquid crystal display (LCD) monitor (MB169B+; ASUS, Taipei, Taiwan) through universal serial bus (USB) type-C connection. Current signals were low-pass filtered at 2 kHz with FL-4 four-pole Bessel filter (Dagan, Minneapolis, MN, USA) to minimize background noise. Through digital to analog conversion, the pCLAMP-generated voltage clamp profiles with various waveforms were specifically designed and suited for evaluating either the relationship between current density and voltage or the steady-state activation curve for the densities in different types of ionic currents such as *I*_h_ [41]. As high-frequency stimuli were needed to elicit the cells, an Astro-med Grass S88X dual output pulse stimulator (Grass Technologies, West Warwick, RI, USA) was used.

### 4.5. Data Analyses

To determine the concentration-dependent inhibition of HNK on the density of *I*_h_ or *I*_K(DR)_, we bathed cells in Ca^2+^-free Tyrode’s solution. To measure *I*_h_ density, we clamped the examined cell at −40 mV and hyperpolarizing pulses up to 2 s in duration to −110 mV were delivered, while, to record *I*_K(DR)_, the examined cell was depolarized from −50 mV for 1 s to +50 mV. The *I*_h_ or *I*_K(DR)_ densities at the end of hyperpolarizing or depolarizing pulses were respectively measured in the control and during the exposure to different HNK concentrations. The concentration required to inhibit 50% of the current density was determined by means of a modified Hill function:(1)$$\mathrm{Relative}\;\mathrm{density}=\frac{{\left[HNK\right]}^{-n_{H}}\times \left(1-a\right)}{{\left[HNK\right]}^{-n_{H}}+IC_{50}^{-n_{H}}}+a$$

In this equation, IC_50_ and n_H_ are the concentration required for 50% inhibition and the Hill coefficient, respectively; [HNK] denotes the honokiol (HNK) concentration applied; maximal inhibition, namely 1-*a*, was estimated from this equation. This formula could reliably converge to give the best-fit line and parameter estimates (e.g., IC_50_ and n_H_).

To characterize the inhibitory action of HNK on *I*_h_ density, we constructed the steady-state activation curve of the current by using a two-step protocol. The relationships between the conditioning pulses and the normalized densities of *I*_h_ with or without the HNK (10 μM) addition were then least-squares fitted by a Boltzmann function in the following form:(2)$$\frac{I\;\mathrm{density}}{I_{\max}\;\mathrm{density}}=\frac{1}{1+e^{\left\{\frac{\left(V-V_{1/2}\right)\mathrm{qF}}{\mathrm{RT}}\right\}}}$$
where *I*_max_ density is the maximal activated density of *I*_h_; V is the conditioning potential in mV; V_1/2_ is the membrane potential (in mV) for half-maximal activation; q is the apparent gating charge; and, F, R and T are the Faraday constant, the universal gas constant and the absolute temperature, respectively.

### 4.6. Statistical Analyses

Linear or nonlinear curve fitting (e.g., sigmoidal or exponential fitting) to data sets was achieved with the least-squares minimization procedure by the use of different maneuvers including the Microsoft Solver function embedded in Excel 2016 (Microsoft) or 64-bit OriginPro^®^ program (OriginLab). The averaged results are presented as the mean ± standard error of the mean (SEM) with sample sizes (*n*) indicating the cell numbers from which the data were collected. The paired or unpaired Student’s *t*-test and a one-way analysis of variance (ANOVA) followed by the post-hoc Fisher’s least-significance difference method, were implemented for the statistical evaluation of differences among means. Assuming that the results obtained did not comply with the assumption of normality, we used the non-parametric Kruskal–Wallis test. Statistical analyses were performed using the SPSS 20 statistical software package (AsiaAnalytics, Taipei, Taiwan). Statistical significance was determined at a *p*-value of < 0.05.

## 5. Conclusions

The notable findings in this investigation are as follows: (a) the presence of HNK produced the concentration-dependent inhibition of *I*_h_ density in GH_3_ cells with an IC_50_ value of 2.1 μM; (b) it shifted the steady-state activation curve of *I*_h_ density toward a more negative potential with no modification in the gating charge of the current; (c) HNK was able to attenuate the voltage-dependent hysteresis of *I*_h_ density elicited by a long-lasting triangular ramp pulse; (d) with the increasing duration of the upsloping ramp pulse upon its return from long-step membrane hyperpolarization, the magnitude of the HNK-induced block, deactivating *I*_h_ density, increased; (e) the HNK addition depressed the density of *I*_K(DR)_ with predicted IC_50_ of 6.8 μM, along with no noticeable change in the activation or inactivation time course of the current density; (f) under cell-attached current recordings, HNK decreased the frequency of spontaneous action currents; (g) finally, this compound was also able to depress the *I*_h_ density identified in Rolf BT.1 olfactory neurons.

## Figures and Tables

**Figure 1 ijms-21-04260-f001:**
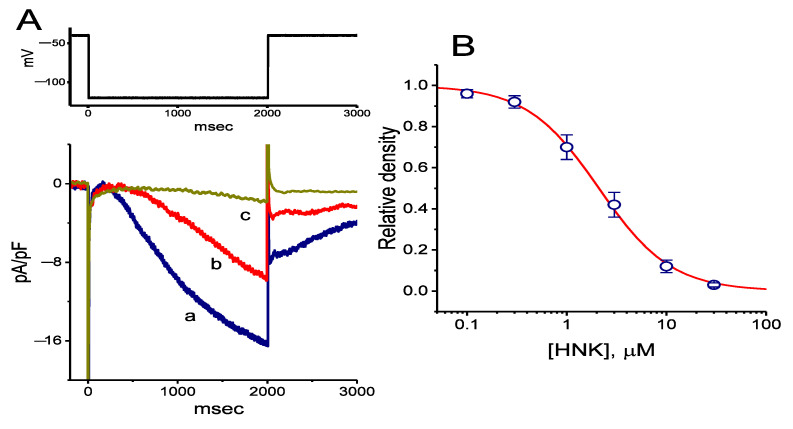
Concentration-dependent inhibition of hyperpolarization-activated cation current (*I*_h_) by honokiol (HNK) in GH_3_ cells. (**A**) Representative *I*_h_ densities obtained in the absence and presence of HNK. In these experiments, we bathed cells in Ca^2+^-free Tyrode’s solution containing 1 μM tetrodotoxin, the recording electrode was filled with a K^+^-containing solution, and the hyperpolarizing pulses from −40 to −120 mV were applied with a duration of 2 s. Current density labeled a is the control, and that labeled b or c was obtained after the addition of 3 μM HNK or 10 μM HNK, respectively. The upper part denotes the voltage clamp protocol applied. (**B**) Concentration–response relationship for HNK-induced inhibition of *I*_h_ density (mean ± standard error of the mean (SEM); *n* = 8 for each data point). The sigmoidal curve represents best fit to the modified Hill equation, and effective half-maximal concentration (IC_50_) value was calculated to be 2.1 μM. (**C**) Summary bar graph showing the effect of HNK on the activation time constant (τ_act_) of *I*_h_ density (mean ± SEM; *n* = 7 for each bar). Current density was obtained by 2-s hyperpolarizing cells to −120 mV from a holding potential of −40 mV. ***** Significantly different from control (*p* < 0.05) and ^†^ significant different from HNK (1 μM) alone group (*p* < 0.05). (D) Mean current density versus voltage relationships of *I*_h_ obtained in the control (□), during exposure to 10 μM HNK (△), and after washout of the compound (○) (mean ± SEM; *n* = 7 for each data point). Current densities were obtained at the end of the 2-s hyperpolarizing steps from a holding potential of −40 mV.

**Figure 2 ijms-21-04260-f002:**
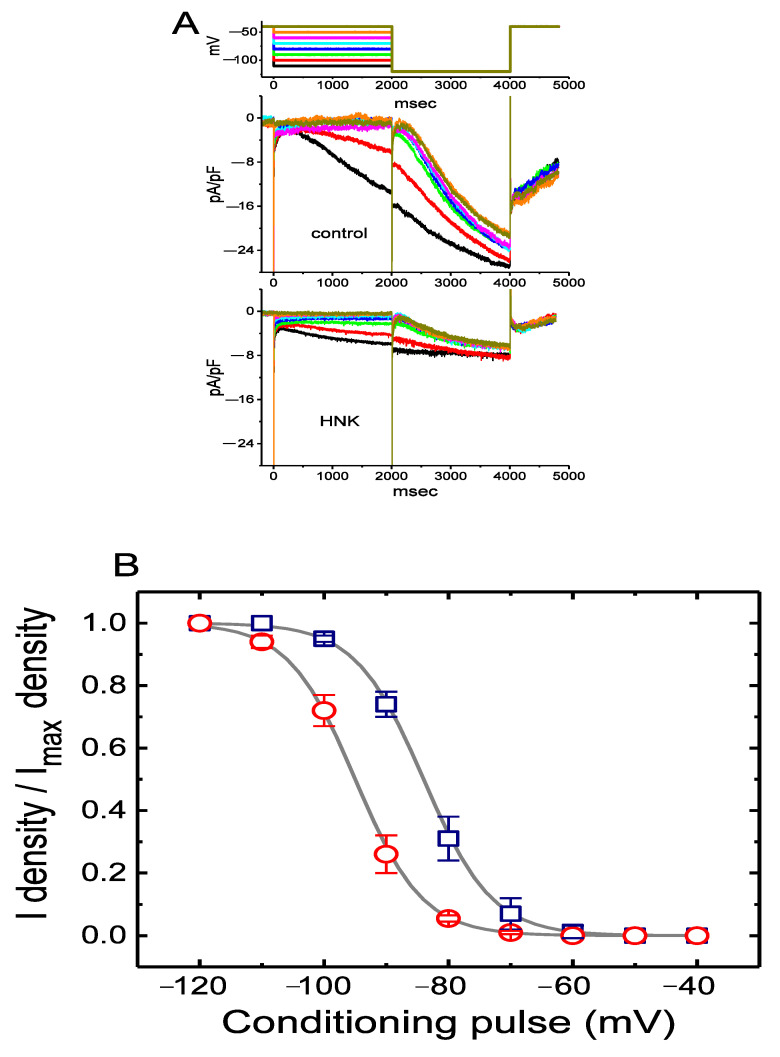
Effect of HNK on the steady-state activation curve of *I*_h_ density measured from GH_3_ cells. These experiments were conducted with a two-step voltage pulse. The conditioning voltage pulses with a duration of 2 s to the potentials ranging from −120 to −40 mV. After each conditioning pulse, a test pulse to −120 mV with a duration of 2 s was delivered to evoke *I*_h_ density. (**A**) Representative *I*_h_ densities elicited in response to the two-step protocol as indicated in the uppermost part. Current densities in the upper or lower panel were obtained in the absence (upper) or presence (lower) of 10 μM HNK, respectively. The uppermost part is the voltage protocol used. (**B**) Steady-state activation curves of *I*_h_ density in the control (□) and during cell exposure to 10 μM HNK (○) (mean ± SEM; *n* = 7 for each data point). Current densities were taken at the end of hyperpolarizing pulse. The sigmoidal smooth curves indicate the least-squares fits to the Boltzmann equation elaborated in **Materials and Methods**. Notably, the presence of HNK decreases *I*_h_ density at more hyperpolarized potentials compared to the control value (i.e., HNK was not present).

**Figure 3 ijms-21-04260-f003:**
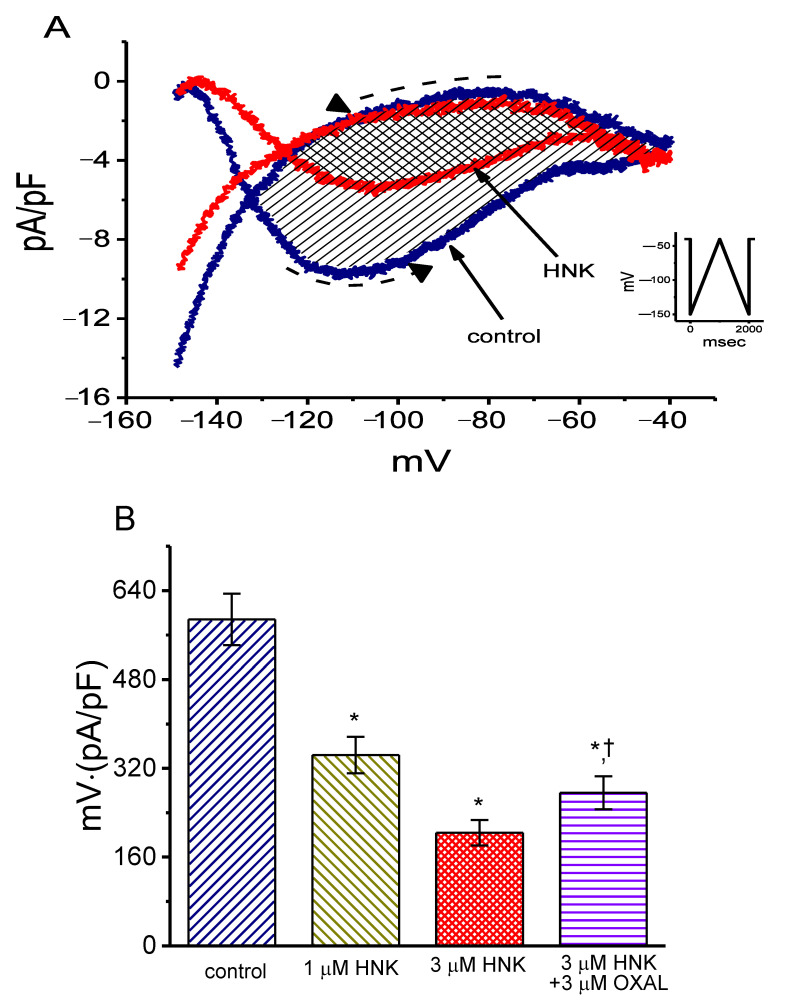
The effect of HNK on the voltage-dependent hysteresis taken from GH_3_ cells. (**A**) Representative *I*_h_ density elicited by 2-s triangular (i.e., upsloping and downsloping) ramp pulse between −150 and −40 mV. Inset in the right lower corner in (**A**) is the voltage protocol applied during the recordings. The voltage hysteresis (i.e., forward or reverse current versus voltage relationship) of *I*_h_ density was detected in the absence and presence of 3 μM HNK. The dashed arrow in (**A**) denotes the direction in which time passes. (**B**) Summary bar graph showing the effect of HNK, HNK and oxaliplatin (OXAL) on the Δarea (as indicated in shaded area in (**A**)) of voltage hysteresis (mean ± SEM; *n* = 8 for each bar). ΔArea taken with or without addition of HNK is indicated as shaded area in (**A**). ∗ Significantly different from control (*p* < 0.05) and ^†^ significantly different from HNK (3 μM) alone group (*p* < 0.05). (**C**) Representative *I*_h_ density in response to 2-s triangular (downsloping and upsloping) ramp pulse obtained in the absence and presence of 3 μM HNK. Inset is the triangular pulse applied, and the dashed arrow indicates the direction in which time passes.

**Figure 4 ijms-21-04260-f004:**
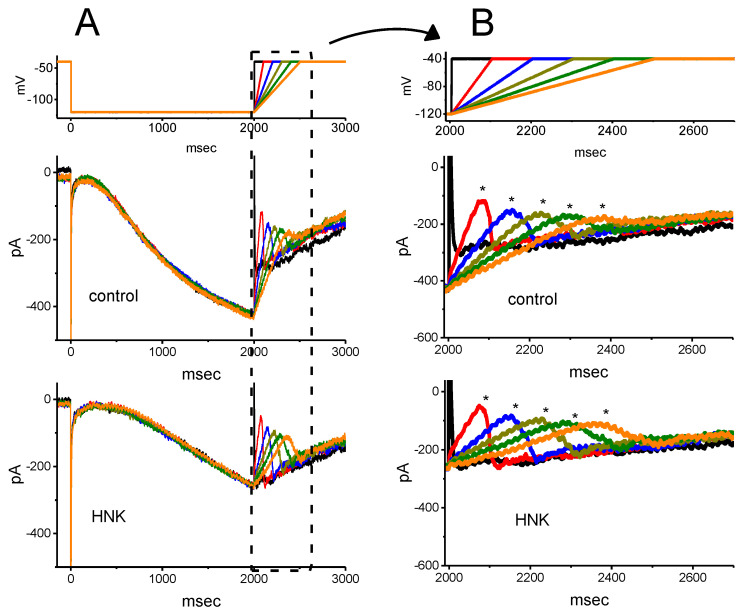
Effect of HNK on the deactivating *I*_h_ density in response to upsloping depolarizing pulses with different durations (100–500 ms) of rising phases, which were used to mimic different rising slopes of pacemaker potential. (**A**) Representative current densities in response to the uppermost voltage protocol obtained in the absence (upper) and presence (lower) of 3 μM HNK. The uppermost part indicates the voltage profile applied. (**B**) Expanded traces of current densities obtained from the dashed box in (**A**). The asterisks shown in (**B**) indicate the hump component of *I*_h_ density in response to abrupt upsloping linear pulse with different durations. (**C**) Effect of HNK on deactivating *I*_h_ density (i.e., the hump component) upon return to −40 mV with different durations (mean ± SEM; *n* = 9 for each data point). The peak density of deactivating *I*_h_ was taken at different durations of rising phase. □: control; ○: in the presence of 3 μM HNK.

**Figure 5 ijms-21-04260-f005:**
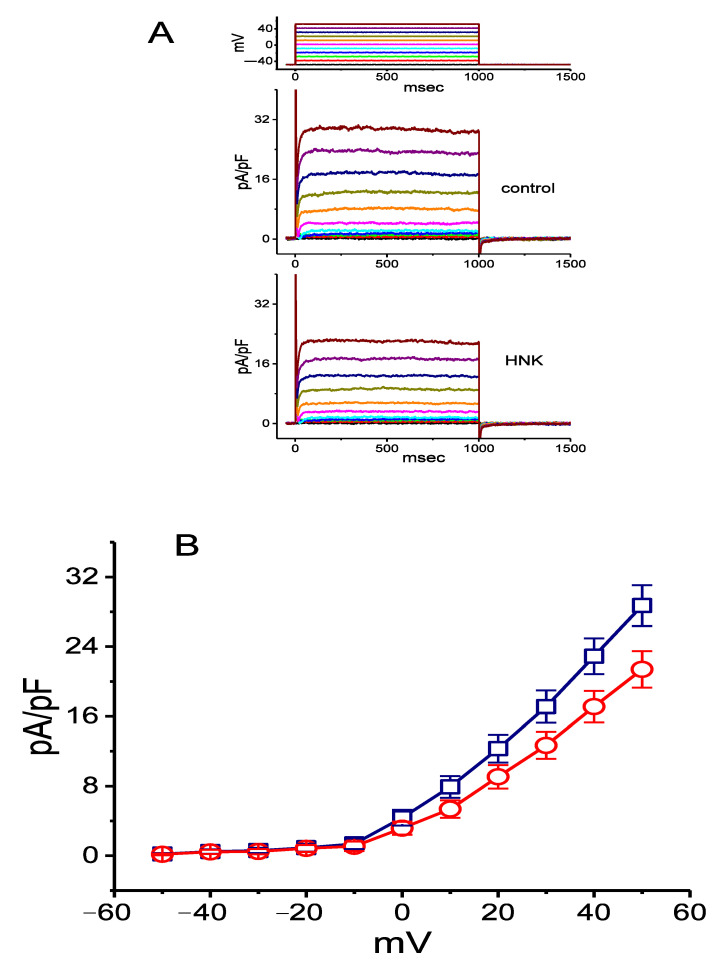
Inhibitory effect of HNK on the density of delayed-rectifier K^+^ current (*I*_K(DR)_) in GH_3_ cells. In these experiments, cells, bathed in Ca^2+^-free Tyrode’s solution containing tetrodotoxin (1 μM) and CdCl_2_ (0.5 mM), were clamped at −50 mV and the voltage pulses from −50 to +50 mV in 10-mV steps were applied. Representative *I*_K(DR)_ densities shown in the upper part of (**A**) are controls, and those in the lower part were recorded 2 min after the application of 10 μM HNK. The uppermost part is the voltage protocol used. (**B**) Mean current density versus voltage relationships of *I*_K(DR)_ density obtained in the absence (□) and presence of 10 μM HNK (○) (mean ± SEM; *n* = 8 for each data point). Current density was measured at the end of each depolarizing pulse. (**C**) Concentration–response relationship for HNK-induced inhibition of *I*_K(DR)_ density (mean ± SEM; *n* = 7 for each data point). Current densities at the end of the pulse were generated as the examined cells were depolarized from −50 to +50 mV with a duration of 1 s. The smooth line represents best fit to the modified Hill equation. The value for IC_50_ was estimated to be 6.8 μM.

**Figure 6 ijms-21-04260-f006:**
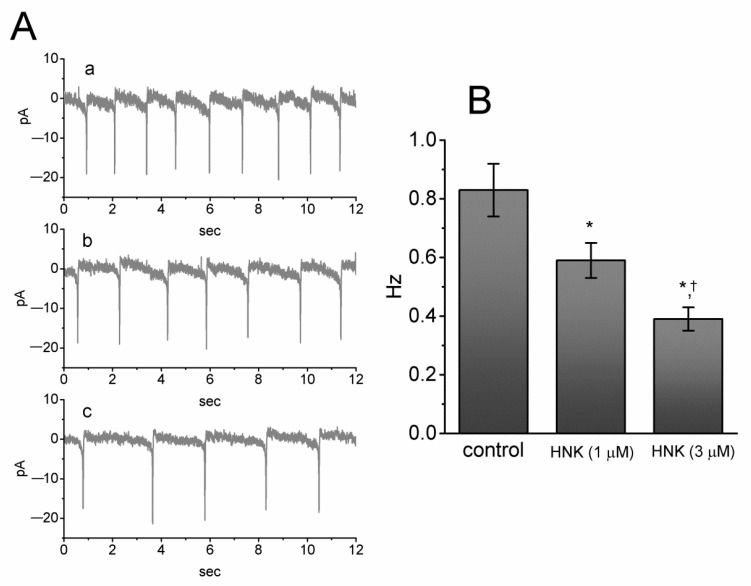
Effect of HNK on spontaneous action currents (ACs) in GH_3_ cells. In this set of experiments, we performed cell-attached current recordings in cells bathed in normal Tyrode’s solution containing 1.8 mM CaCl_2_ and the potential was held at the resting potential of the cells (around −70 mV). (**A**) Representative ACs taken from the absence or presence of HNK. Current trace labeled a is control, and that labeled b or c was obtained 2 min after addition of 1 μM HNK or 3 μM HNK, respectively. The downward deflections represent the capacitive currents that charge the surface membrane. (**B**) Summary bar graph depicting effect of HNK on the firing frequency of spontaneous ACs. Each bar indicates the mean ± SEM (*n* = 8). * Significantly different from control (*p* < 0.05) and ^†^ significantly different from HNK (1 μM) alone group (*p* < 0.05).

**Figure 7 ijms-21-04260-f007:**
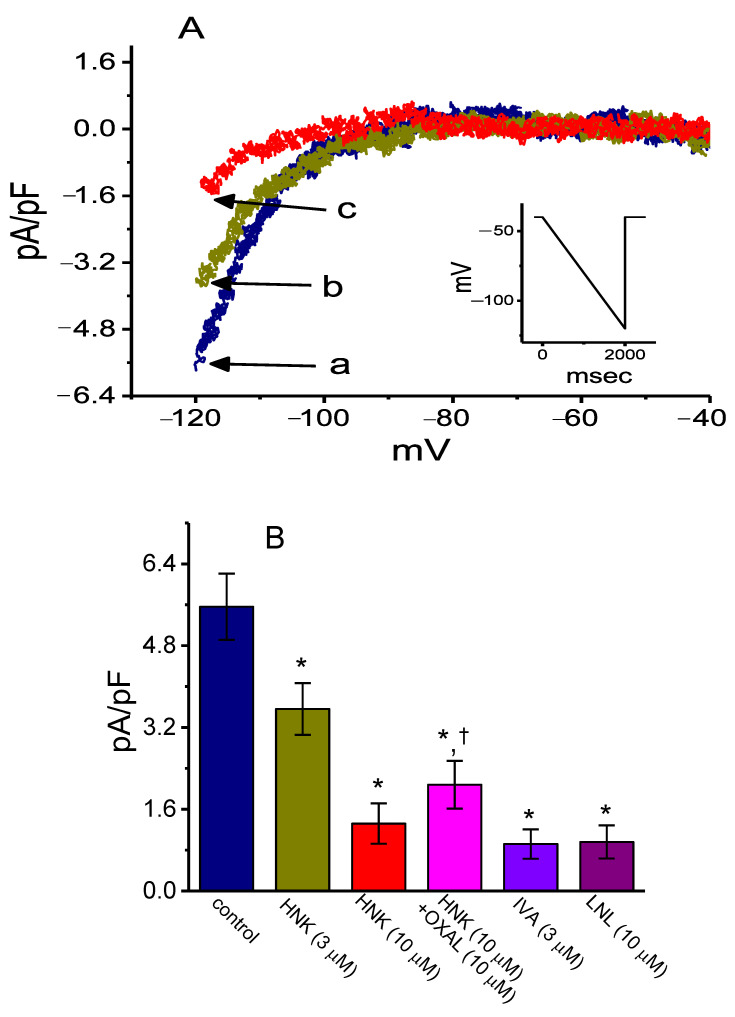
Effect of HNK on *I*_h_ density elicited by a downsloping ramp pulse in Rolf 1B olfactory neurons. In this set of experiments, we voltage clamped the examined cells at −40 mV and applied the linear ramp pulse from −40 to −120 mV with a duration of 1 s (as indicated in the inset of (**A**)). (**A**) Representative *I*_h_ density elicited by such linear ramp taken from the absence or presence of HNK. Current density labeled a is control, and that labeled b or c was obtained 2 min after addition of 3 μM HNK or 10 μM HNK, respectively. (**B**) Summary bar graph depicting effect of HNK, HNK plus oxaliplatin (OXAL), ivabradine (IVA), or linalool (LNL) on *I*_h_ density in response to ramp pulse. Current density was measured at the level of −120 mV. Each bar indicates the mean ± SEM (*n* = 8). * Significantly different from control (*p* < 0.05) and ^†^ significantly different from HNK (10 μM) alone group (*p* < 0.05).

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
