# Peer review of "Effectiveness in the Block by Honokiol, a Dimerized Allylphenol from *Magnolia Officinalis*, of Hyperpolarization-Activated Cation Current and Delayed-Rectifier K^+^ Current"

_ijms, 2020, doi:10.3390/ijms21124260_

Round 1

Reviewer 1 Report

I have enjoyed reading the results of your study on the effects of honokiol (HNK) on selected ion channels studied using voltage clamp methods combined with selected over expression in cultured cell systems.  I agree with your major conclusion, that HNK at low micro molar levels can block the hyperpolarization-activated current, IH.  However, the data concerning block of the delayed rectifier current and the effects of HNK in olfactory sensory neurons are much less convincing, in part because of the limited data sets that are presented.  My suggestions for clarifying and improving your manuscript are:

1.  As mentioned, your data sets quite convincingly demonstrate an inhibitory effect of HNK on IH.  This finding could be strengthened substantially if additional data was included that demonstrated this effect was reversible.  In the absence of this, particularly when IH channels are over expressed in a cell line some of the biophysical effects that you observed could be 'voltage clamp artefacts'.

2.  With respect to your analysis of IH most of your experimental design is conventional and convincing.  However, when you attempt to present and discuss the kinetic effects that form the basis of Figures 3 and 4 additional clarification is needed.  The hysteresis that you draw attention to is generated mainly by what I would term 'residual activation'.  That is, the deactivation tails have a very slow time course.  I would prefer drawing attention to this and also commenting on the marked sigmoidicity of the activation phase of this current and the biphasic tail records as opposed to your indirect characterization using ramp voltage clamp command waveforms.  Second, and importantly, all of your data should be presented in terms of current density (pA/pF) so that the likelihood of its physiological relevance could be assessed.  Importantly also, this will provide information concerning whether or not the IH activation curves that you present can be described as 'steady-state', or whether they are 'isochronal'.  This distinction is important if you are going to use the maximum slope of any of these relationships to learn about gating charge (see line 24).

3.  Regarding possible physiological relevance, the voltage ranges in which IH activation occurs either in control conditions or after HNK treatment seem to fall outside of the values of membrane potential in a neuron.  Please comments.

4.  Overall, your work is thorough and presented in a logical fashion but this does not apply to the last sections of the Results namely Figures 5 and 6.  In both cases the data sets are too small and the lack of data showing reversibility detract from an ability to interpret these findings.

5.  Your manuscript is readable but could be improved with additional editing to improve Scientific English and clarity.  Examples in the Discussion section include:

a)  line 415 - 'became raised'

b)  line 430 - 'pose excitable'

c)  line 442 - 'growingly'

d)  line 465 - 'voltage-dependent elicitation'

e)  line 475 - 'a current study'

f)  line 483 - 'in the end', and 'unravels'

Author Response

We are grateful of the reviewer’s comments on ways to improve our work. The reply below follows the order of specific questions raised by the reviewer.

  1. Thanks for the comments by the reviewer, stating that the experimental results of ours are convincing. In the present study, the experimental results were not deemed to be overexpressed, since either Ih or IK(DR) is functionally expressed in pituitary GH3 cells. The washout data were included in the revised manuscript (lines 146-147, 290-291, and 341). Additionally, we believed that the data demonstrated in this work would not be “voltage-clamp artefacts”.
  2. Thanks for the critical comments raised by the reviewer. An additional paragraph (lines 422-426) regarding “residual activation” was included in the Discussion section of the revised manuscript. Moreover, as per the suggestion by the reviewer, the Figures were significantly redone in the revised manuscript. Please note that current amplitudes in Figures were totally replaced with current densities and the texts in the manuscript were correspondingly changed. Additionally, in our study, the presence of honokiol (HNK) was observed not only to reduce the maximal conductance of Ih, but also to significantly shift the activation curve in parallel to a hyperpolarized potential by approximately 11 mV.
  3. Although the voltage ranges in which Ih activation occurs either in control conditions or after HNK treatment appear to fall outside of the values of membrane in a neuron, a small fraction of Ih are tonically activated at rest (Robinson and Siegelbaum, Annu Rev Physiol 2003;65:453-480). Hence, the issue with an additional reference was included in the revised manuscript (lines 419-421).
  4. We agree with the comments indicated by the reviewer. The results regarding the washout of the HNK treatment were included in the revised manuscript (lines 146-147, 290-291 and 341).
  5. Thanks for the advice raised by the reviewer.
  6. “became raised” was changed to “was increased” (line 559, in the revised manuscript)
  7. “pose excitable” was appropriately changed to “render excitable cells” (line 374, in the revised manuscript).
  8. “growingly” was removed from the text in the revised manuscript (line 393)
  9. “voltage-dependent elicitation” was changed to “voltage-dependent activation” (line 416, in the revised manuscript).
  10. “a current study” was changed to “a recent study” (line 434, in the revised manuscript).

“in the end” was changed “collectively”, and “unravels” was changed to “shows” (line 444, in the revised manuscript).

Reviewer 2 Report

In the manuscript prepared by Chan et al., the authors investigated the effects of honokiol (HNK) on the hyperpolarization-activated cation current and delayed-rectifier potassium current. They found HNK can block both currents as well as the deactivating current of Ih. In addition, the drug can shrink the hysteresis effect during membrane depolarization and hyperpolarization. Since the hyperpolarization-activated cation current is physiologically very important for establishment of action potential and pace making activities, the findings made here are insightful and potentially can be applied to therapeutics for certain neurological diseases in the future. Here are my suggestions for the authors to revise their manuscript.

  1. In section 3.1 where authors measured the block of hyperpolarization-activated current by HNK, they should also take a look at the activation rate in the absence/presence of HNK. The time course change may yield some important information on how this compound can be used in vitro. A panel showing concentration vs. rate will be helpful.
  2. In section 3.2, as the hyperpolarization-activated current is a mixture of several different channel currents, what is the meaning of the so called ‘gating charge’ that authors repetitively used in the manuscript? The ‘gating charge’ not changing here means what? In the discussion section, the author attributed the main player for the Ih to HCN channels. But according to Baron Chanda et al., 2019 (elife), the gating charge of HCN is estimated to 7.8 e (also refer to other references cited there), which suggests the gating charge authors measured is something else, rather than any specific gating charge of any channel. I think the author want to reconsider how to use this more rigorously.
  3. For the hysteresis experiments. What is the physiological significance of the shaded area (the area between upsloping and downsloping curve)? What is the interpretation on the area change by HNK? Illuminating these points will significantly boost the value of their findings. Otherwise, why bother to study the change? Also the authors should specify how this area is measured in the manuscript.
  4. HCN channels are indeed the main players of Ih, It is possible that HNK affects function of this channel family. Is there a way the authors can investigate this experimentally as this will provide great insights into the drug working mechanism? For example, do a few measurements with the drug and HCN channels?

Author Response

Thanks for the insightful comments provided by the reviewer. The reply shown below follows the order of the comments indicated by the reviewer.

  • As per the reviewer’s advice, an additional panel (i.e., Figure 1C) regarding the activation time constant (i.e., tact) of Ih density elicited by membrane hyperpolarization was included in the revised manuscript (lines 116-117). The text in the revised manuscript was also included (lines114-116). The Figure legend was included as well (line 127-131).
  • Thanks for the comment raised by the reviewer. Hence, since this is potentially important, an additional paragraph regarding this issue (i.e., estimate of the gating charge) was included in the Discussion section of the revised manuscript (lines 386-391). An additional reference (i.e., Kasimova et al., 2019) was incorporated into the References section as well.
  • Voltage-dependent hysteresis of Ih density has been demonstrated to serve an important role in influencing the electrical behavior of electrically excitable cells (Männikkö et al., 2005; Fürst and D'Avanzo, 2015; Barthel et al., 2016). The Ih density inherently in neurons or endocrine and neuroendocrine cells was significantly described either to undergo either a hysteretic perturbation in its voltage dependence, or a significant mode shift in which the voltage sensitivity in gating charge movements of the current depends on the previous state of the HCN channel. Moreover, in this study, we examined the possible modifications of HNK on such non-equilibrium property of Ih density inherently in GH3 Specifically, our results demonstrated that the presence of this compound was apparently capable of diminishing such hysteresis involved in the voltage-dependent activation of Ih density. Moreover, in our work, we further quantified the degree of voltage-dependent hysteresis of Ih by calculating the difference in the area under the curves (indicated in the shaded area) in the upsloping and downsloping directions of long-lasting triangular voltage ramp. Collectively, the presence of HNK shown herein could significantly decrease the area of voltage-dependent hysteresis, strongly reflecting that the amount of hysteresis was diminished in the presence of HNK.
  • The Ih has been regarded to be carried by channels of the hyperpolarization-activated cyclic nucleotide-gated (HCN1-4) gene family, which belongs to the superfamily of voltage-gated K+ channels and cyclic nucleotide-gated channels. It has been described that the HCN2, HCN3, or mixed HCN2+HCN3 channels are functionally expressed in different types of endocrine cells including GH3 cells (Kretschmannova et al., 2012; Spinelli et al., 2018). To what extent HNK could produce differential inhibition on these different HCN-channel currents is important and remains to be further determined. In other words, further investigations are needed to find out whether HNK can affect either Ih existing in a variety of cells or different types of Ih (e.g., HCNx-encoded currents) (lines 407-408 in the revised manuscript).

Reviewer 3 Report

In this manuscript, Chan et al. aimed at unveiling the effects of honokiol (HNK) on the Ih and IK(DR) in pituitary cells (GH3) and olfactory neurons (Rolf B1. T cells). The authors demonstrated that HNK inhibited the amplitude of Ih activated by hyperpolarization in a concentration-, time-, and voltage-dependent manner in GH3 cells. The application of HNK decreased the voltage-dependent hysteresis of Ih elicited by triangular ramp pulse and inhibited the amplitude of IK(DR) in GH3 cells. Moreover, Ih in the Rolf B1. T cells was also inhibited by HNK in a concentration-dependent matter.

Overall the hypothesis is interesting, and the data are of good quality, and this study is very significant. However, I think that this manuscript is not recommended for publication in its present form. It is acceptable for publication in the International Journal of Molecular Sciences (IJMS) with the following corrections (see comment):

Major

  1. Effect of HNK on frequency of spontaneous action potentials (APs) in GH3 cells.

In this manuscript, authors demonstrated that HNK suppresses Ih in GH3 cells. This finding is very important, but I think that more important is the effect of HNK on the frequency of spontaneous APs in GH3 cells. The authors have shown that the application of HNK suppresses both Ih and IK(DR) in BH3 cells but have not examined how HNK alters the excitability of BH3 cells. The effects of HNK on the frequency of action potentials in BH3 cells needs to be clarified.

  1. Physiological role of Ih in pituitary cells.

In this manuscript, little is mentioned about the physiological role of both Ih and IK(DR) in pituitary cells. The authors should describe the physiological role of both Ih and IK(DR) in Discussion section. Moreover, it should also be described how inhibition of both Ih and IK(DR) alters cell function of pituitary cells.

  1. linalool.

There is little description about linalool. I don't know why the authors used this compound. Authors should elaborate on this point.

Minor

  1. Page 1, line 29 – “increase” should read as “decrease”.

Author Response

We appreciate the reviewer’s comments on ways to improve our work.  The reply shown below follows the order of the comments pointed out by the reviewer.

Major:

  1. As per the suggestion by the reviewer, additional set of cell-attached current measurements were conducted. The experimental results along with the relevant text (lines 317-323) were included and an additional Figure (i.e., Figure 6) was accordingly incorporated in the revised version of the manuscript. The experimental data showed the effectiveness of HNK in decreasing the frequency of spontaneous action currents present in pituitary GH3

  1. Physiological role of Ih in pituitary cells

        With regard to the physiological role of Ih, the text related to its physiological role was included in the revised manuscript. That is, “Owing to the pacemaker current, the activation of this current can have the propensity to cause the resting potential to be depolarized and then to reach threshold required for the generation of action potential; as a result, it is thereby able to influence pacemaker activity and impulse propagation” (lines 73-78). Additionally, in terms of IK(DR), the text relevant to its physiological role was included in the Results section of the revised manuscript. That is, “The major determinants of IK(DR) in GH3 cells are the voltage-gated K+ channels from KV3.1-KV3.2 types and was thought to play a role in determining membrane excitability and the repolarization of action potential” (lines 280-282).

  1. Linalool

Linalool, an aromatic plant-derived monoterpene alcohol, has been demonstrated to exert anti-inflammatory and anti-oxidant properties (Sabogal-Guáqueta et al., 2019). Previous studies have demonstrated that linalool or other essential oils (e.g., geraniol, D-limonene) could effectively suppress the firing of action potentials and pentylenetetrazole-induced epileptiform activity (Souto-Maior et al., 2017; Vatanparast et al., 2017; Bahr et al., 2019; Lei et al., 2019). The linalool-induced inhibition of Ih in Rolf B1.T olfactory neurons is likely responsible for its effect on pain sensation (lines 444-448 in the revised manuscript).

References:

Bahr TA, Rodriguez D, Beaumont C, Allred K. The effects of various essential oils on epilepsy and acute seizure: a systematic review. Evid Based Complement Alternat Med 2019;2019:6216745.

Lei Y, Fu P, Jun X, Cheng P. Pharmacological properties of geraniol - a review. Planta Med 2019;85:48-55.

Sabogal-Guáqueta AM, Hobbie F, Keerthi A, Oun A, Kortholt A, Boddeke E, Dolga A. Linalool attenuates oxidative stress and mitochondria dysfunction mediated by glutamate and NMDA toxicity. Biomed Pharmacother 2019;118:109295.

Souto-Maior FN, Fonsêca DV, Salgado PR, Monte LO, de Sousa DP, de Almeida RN. Antinociceptive and anticonvulsant effects of the monoterpene linalool oxide. Pharm Biol 2017;55:63-67.

Vatanparast J, Bazleh S, Janahmadi M. The effects of linalool on the excitability of central neurons of snail Caucasotachea atrolabiata. Comp Biochem Physiol C Toxicol Pharmacol 2017;192:33-39.

Minor:

  • Goof! We made a mistake. “increase” was corrected to “decrease” (line 31).

Round 2

Reviewer 1 Report

Thank you for your clear and quite convincing responses to my questions and suggestions for clarification and improvement of the original manuscript.

Reviewer 2 Report

It seems in line 322, there is a mistake that the 'HCN' should be 'HNK'.

Reviewer 3 Report

In this manuscript, Chan et al. aimed at unveiling the effects of honokiol (HNK) on the Ih and IK(DR) in pituitary cells (GH3) and olfactory neurons (Rolf B1. T cells). The authors demonstrated that HNK inhibited the amplitude of Ih activated by hyperpolarization in a concentration-, time-, and voltage-dependent manner in GH3 cells. The application of HNK decreased the voltage-dependent hysteresis of Ih elicited by triangular ramp pulse and inhibited the amplitude of IK(DR) in GH3 cells. Moreover, Ih in the Rolf B1. T cells was also inhibited by HNK in a concentration-dependent matter.

Overall the hypothesis is interesting, and the data are of good quality. I am satisfied with the additional experiments and revised parts. I think that the revised manuscript is acceptable for publication.